paediatrics; therapeutics; precision medicine; pharmacogenomics; drug safety

**Author for correspondence:**
Michael Rieder,
Email: mrieder@uwo.ca

# Advancing Precision Medicine in Paediatrics: Past, present and future

Abdelbaset Elzagallaai[1], Charlotte Barker[2], Tamorah Lewis[3], Ronald Cohn[3] and Michael Rieder[4] 

[1]Department of Physiology and Pharmacology, University of Western Ontario, London, ON, Canada; [2]Department of Medical and Molecular Genetics, King's College London, London, UK; [3]Department of Paediatrics, Hospital for Sick Children, University of Toronto, Toronto, ON, Canada and [4]Departments of Paediatrics, Physiology & Pharmacology and Medicine, University of Western Ontario, London, ON, Canada

## Abstract

Precision Medicine is an approach to disease treatment and prevention taking into account individual genetic, environmental, therapeutic and lifestyle variability for each person. This holistic approach to therapeutics is intended to enhance drug efficacy and safety not only across healthcare systems but for individual patients. While weight and to some extent gestational age have been considered in determining drug dosing in children, historically other factors including genetic variability have not been factored into therapeutic decision making. As our knowledge of the role of ontogeny and genetics in determining drug efficacy and safety has expanded, these insights have provided new opportunities to apply principles of Precision Medicine to the care of infants, children and youth. These opportunities are most likely to be achieved first in select sub-groups of children. While there are many challenges to the successful implementation of Precision Medicine in children including the need to ensure that Precision Medicine enhances rather than reduces equity in children's health care rather, there are many more opportunities. Research, advocacy, planning and teamwork are required to move Precision Medicine forward in children in pursuit of the common goal of safe and effective drug therapy.

## Impact statement

Precision Medicine – the integration of many sources of variability in patient response, including genetic and non-genetic sources, is a therapeutic approach that seems here to stay. The rapid advance of the new biology and the Second Therapeutic Revolution will make Precision Medicine and the use of genetic information and molecular targeting more and more relevant in the therapeutic space. While the implementation of Precision Medicine is well underway for adult patients, progress in Paediatrics has been slower, despite emerging evidence that Precision Medicine approaches can make drug therapy for children both more effective and safer. Challenges to successful implementation include educational, organisational, leadership and health system issues including integration of health care delivery and overcoming established and artificial disciplinary barriers. To overcome these challenges, advocacy and a multidisciplinary approach are needed. It is likely that the best way forward is the targeted implementation of Precision Medicine in focused areas such as oncology, neonatology and for children with complex chronic disease. While this is proceeding, it is also important to recognise that Precision Medicine has the potential to increase inequity in health care for children, a key consideration given the already recognised inequity in the delivery of optimal therapeutics in Paediatrics. The successful implementation of Precision Medicine in Paediatrics offers considerable potential to enhance our ability to provide effective and safe drug therapy for the world's children.

## Introduction: Drug use in children

A child born in the United Kingdom – or for that matter most of the developed world – is expected to live for approximately 81 years. It has not always been so – in fact, in 1920 the life expectancy at birth was in the range of 58 years. In 1920, the under-five mortality rate for children in the United Kingdom was 14% – roughly one in every six children died before their fifth birthday (https://www.statista.com/statistics/1041714/united-kingdom-all-time-child-mortality-rate/; Bideau et al., 1997). Currently, the under-fiver mortality rate in the UK is 0.4%, primarily related to issues around delivery and prematurity.

This was primarily due to three things – public sanitation, vaccination and effective drug therapy (Thomas, 1983; Weinshilboum, 1987; Bonanni, 1999; Rieder, 2010). While the impact of public sanitation and vaccination cannot be overstated the impact of drug therapy was extremely dramatic, in part because it occurred so rapidly (Rieder, 2010). While Sir Alexander Fleming

described the anti-bacterial effects of penicillin in 1929, there was initially little interest in its use as an antimicrobial agent, in part due to difficulties in securing large quantities of the drug (Fleming, 1929). In contrast, when Gerhard Domagk discovered as part of work with dyes that Prontosil cured *Streptococcal* infections in mice in 1935, this was rapidly translated into therapy; by 1937 sulfanilamide (Prontosil a pro-drug metabolised to sulfanilamide) was in widespread clinical use, and is credited for saving thousands of lives (Weinshilboum, 1987).

While the Therapeutic Revolution transformed the care of children, it was not without cost. The Elixir of Sulfanilamide Tragedy highlighted the need for robust pre-clinical testing while the Thalidomide Disaster and the Chloramphenicol Grey Baby Syndrome demonstrated the vulnerability of neonates and the need to appreciate how ontogeny impacted on drug efficacy and safety (McBride, 1961; Lietman, 1979; Wax, 1995). The regulatory changes triggered by these events – notably the Kefauver–Harris amendments to the Food and Drug Act in 1962 – inadvertently created an environment when drug research in children was discouraged, resulting in Harry Shirkey's famous characterisation of children as "therapeutic orphans" (Shirkey, 1968; Done et al., 1977). Advocacy over time and regulatory changes as well as shifts in ethical outlook on the inclusion of children in research did eventually result in a renewed interest in drug research in children, with impacts that have profoundly affected care (Kauffman, 1995; Wilson, 1999; Moore-Hepburn and Rieder, 2021).

It is important to appreciate that drug therapy in children is much more common than previously appreciated. When we studied drug use among a million Canadian children, we found that on average they received four prescriptions a year (Rieder et al., 2003). This figure is somewhat deceptive; most children received either no prescriptions in a year or a single prescription, often an antibiotic. Conversely, 20% of the children received 70% of the prescriptions; these were typically children with complex and/or chronic diseases, who received therapy from a very wide range of medications (Rieder et al., 2003). Subsequent work in other countries has demonstrated similar findings; medications are used in children much more than commonly believed and come from a very wide range of therapeutic classes (Clavenna and Bonati, 2009; Hales et al., 2018).

## Sources of variability in drug response in children

Dr. Abraham Jacobi, a key pioneer in American paediatrics, famously noted that "Pediatrics does not deal with miniature men and women, with reduced doses and the same classes of diseases in smaller bodies, but …. it has its own independent range and horizon" (Jacobi, 1889; Gillis and Loughlan, 2007). This is especially true for therapeutics, notably as failing to appreciate this has resulted in tragedy (Lietman, 1979; Choonara and Rieder, 2004). A major source of variability in drug response is ontogeny. Over the course of childhood from the time of birth to late adolescence, there is as much a 50-fold difference in drug disposition, most pronounced in the first year of life (Kearns et al., 2003; Chapron et al., 2021). There has been considerable progress over the past two decades in understanding how changes in absorption, distribution, metabolism and excretion of drugs changes over the course of development, providing considerable understanding of how changes in drug clearance impact therapy, notably with respect to drug safety (Chapron et al., 2021). Over the past decade, it has also become apparent that changes in drug disposition at the level of

tissues and cells with respect to drug transporter expression and function can affect drug efficacy and safety (van Groen et al., 2020; van Groen et al. 2021). While body weight as a determinant of dose for children has been used clinically for many years, it is now clear that allometric approaches to determine drug dosing need to include a fulsome consideration of variables including age, distribution (including plasma protein binding), route of elimination route, isoenzyme maturation, renal development and transporter biology (Hu, 2022; van Rongen et al., 2022).

In addition to changes related to ontogeny, drug disposition and effects can also be impacted by concurrent therapy. Polypharmacy is common among children with complex and chronic conditions (Rieder et al., 2003; Clavenna and Bonati, 2009; Hales et al., 2018; Fraser et al., 2022). While the importance of appreciating interactions in the context of polypharmacy has been appreciated for some time by physicians caring for adult patients, this has been less commonly appreciated by providers of health care to children (Anand et al., 2021). Drug–drug interactions can also involve traditional or folk medications, which can be used more often than expected in children with complex and chronic disease, as well as drugs that depending on the jurisdiction may or may not be illicit such as cannabidiol. As well, while not germane for all drugs diet and exercise can impact on drug efficacy and safety; there is a lack of understanding on how this may impact the care of children (Niederberger and Parnham, 2021). Disease can also affect drug response and efficacy, notably in the case of certain viral infections (van Tongeren et al., 2020).

Finally, while historically underappreciated, the role of genetics in determining variability in drug efficacy and response cannot be understated.

## Genetic determinants of drug efficacy and safety in children

The importance of genetics is something that has never been lost to paediatricians. Sir Archibald Garrott noted early in the 20[th] century that it was likely that genetics had a major role in determining variability in drug action (van Tongeren et al., 2020; Vernon and Manoli, 2021). However despite the appreciation that genetics was likely important in human health and the description of the genetic inheritance of a number of disorders, for many decades genetics was primarily focused on basic science and statistical approaches (Rimoin and Hirschhorn, 2004). It was only in 1948 that the American Society of Medical Genetics was formed; while many pioneers in human genetics were focused on adult Internal Medicine, very shortly a number of paediatricians began work in this area, notably Dr. Cedrick Carter and colleagues at The Hospital for Sick Children at Great Ormond Street (Rimoin and Hirschhorn, 2004).

Pharmacogenomics arose as a result of the work of key pioneers, two geneticists – Drs. Arno Motulsky in the United States and Friedrich Vogel in Germany and a pharmacologist, Dr. Werner Kalow in Canada; Motulsky's classical paper in 1957 recognised how genetics, biochemistry and pharmacology could interact to determine a drug's effects, while Vogel is credited with coining the term pharmacogenetics in 1959 and Kalow published the first book presenting a systematic approach to pharmacogenetics in 1962 (Motulsky, 1957; Vogel, 1959; Kalow, 1962; Rimoin and Hirschhorn, 2004; Blankstein, 2014). Initially considered an arcane academic pursuit, expanding capacity in biochemical, pharmacological and genetic research has over the past six decades, with discoveries such as the polymorphic nature of CYP2D6 and TMPT (Taylor

et al., 2020; Pratt et al., 2022). These discoveries have been translated into clinical care for adult patients, with more than 260 drugs currently having a pharmacogenomic label in their FDA product monograph. In many healthcare systems, pharmacogenomic testing is part of routine for medications primarily directed at adults such as warfarin (Eissenberg and Aurora, 2019; Asiimwe and Pirmohamed, 2022).

Historically pharmacogenomic approaches have been very targeted. Evolving capacity in genomic analysis may change this. Genome sequencing is a comprehensive test capable of detecting nearly all DNA variation in a genome and is a significant component of the Precision Medicine concept. Sequencing can identify most of the 7,000 diseases mentioned in the Online Mendelian Inheritance in Man database (www.omim.org), which has a known genetic basis (www.omim.org). These include cystic fibrosis, Duchenne muscular dystrophy, familial hypercholesterolemia and haemophilia. Patients may present with unusual constellations of features, or with common diseases like autism spectrum disorder, cardiomyopathy, congenital heart disease, epilepsy, cancer, schizophrenia or dementia, although this list is not comprehensive. Genome sequencing is broader in scope than other commonly used genetic tests, and data can be analysed in both hypothesis-driven and hypothesis-generating ways. For these reasons, genome sequencing will most certainly eclipse exome sequencing, large next-generation sequencing gene panel tests, and chromosomal microarray analysis in the future.

Genome sequencing is a three-stage process. First, after obtaining informed consent a medical geneticist or other health care professional obtains the required information on the patient's phenotype and family history. Second, a clinical laboratory geneticist analyzes the genome data. Third, a physician compares the genetic findings to the clinical manifestation to assess risk. The overall goal of interpreting a genetic variant is to explain it in the context of all or part of the clinical manifestation. The main aim of genome sequencing as a clinical diagnostic test is to identify these variations. Some laboratories in North America will also search for secondary findings, which are disease-causing variants in genes associated with medically actionable conditions that are unrelated to the initial intent for testing.

The procedure of sequencing is safe; however, possible negative consequences are tied to how results are interpreted and disclosed. First, genome sequencing may be misinterpreted as a diagnostic that may answer all clinical questions. Clarity in clinical data and family history is still essential for interpreting findings. A positive result does not necessarily explain all of the patient's characteristics, and a negative test does not indicate that there was no genetic component or invalidate an obvious clinical diagnosis. Second, because of ongoing understanding and the characterisation of new information, the classification of a genetic variant may change over time. The majority of ancient peoples other than Europeans are under-represented in the big-scale reference databases of genomic variation that guide interpretation, thus misdiagnosis is a risk for these people that needs to be taken into consideration. Third, genetic test results might provide information about the person, his family members, or their connections to one another that was not previously considered. These facts underscore the need for thorough pre- and post-test counselling and qualified genetics professionals.

While there has been dramatic progress in moving pharmacogenomic testing into the realm of adult medicine, this has been less so in paediatrics, for many of the reasons detailed below. However, in certain key groups of children, pharmacogenomic testing has been moving rapidly over the past decade.

## Oncology

Cancer is an important cause of morbidity and mortality in children globally; it is the leading cause of death in American children over the age of 1, and while progress continues in reducing mortality rates for hematogenous and central nervous system cancers survival rates for bone and soft tissue cancers have plateaued (Siegal et al., 2020; Huang et al., 2022). Given the centrality of chemotherapy in cancer therapy and the complexity of chemotherapy metabolism, better understanding not only of issues such as drug–drug interactions but also how genetic variations influence drug safety and efficacy is of clear and immediate importance (Elzagallaai et al., 2021).

The demonstration that thiopurine methyl transferase (TPMT), an enzyme central to the metabolism of the very commonly used chemotherapeutic drug 6-mercaptopurine (6-MP), exists as a monogenic autosomal codominant trait, with the number of loss-of-function alleles determining if a patient is a normal, intermediate or low metabolizer which in turn predicts if a patient can tolerate conventional dosing without a sharp increase in the risk of myelosuppression was a crucial step in bringing pharmacogenomics into cancer care (Relling et al., 1999). These insights have been translated into clinical guidelines and have in many healthcare systems mandated TPMT genotyping for children with cancer being treated with 6-MP – and subsequent dose adjustment – as part of their therapeutic protocols (Relling et al., 2019).

In the case of oncology, much of the work in understanding the impact of genetic variations has been to reduce the risk of adverse drug reactions (ADRs) – a major issue during therapy – while maintaining therapeutic efficacy (Elzagallaai et al., 2021). Variations in key DNA repair pathways have been demonstrated to be associated with resistance to platinum-based chemotherapy as well as in tolerability to therapy (Sakano et al., 2010; Zheng et al., 2017). Variations in other genes – including TPMT – have been associated with increased risk for hearing loss in children treated with platinum-based chemotherapy, a significant ADR associated with these treatments (Ross et al., 2009; Thiesen et al., 2017; Drögemöller et al., 2018; Clemens et al., 2019). However, this association has been a matter of debate as other studies were not able to replicate the findings (Langer et al., 2020).

The anthracyclines are a major class of chemotherapeutic agents central in the therapy of leukaemias and lymphomas, among the commonest cancers in children. Although of unquestioned efficacy, they are associated with numerous serious ADRS including myelosuppression and the risk of cardiomyopathy and cardiac failure (Peng et al., 2005; Geisberg and Sawyer, 2010). Exploration of possible genetic variations that determine risk has shown that a series of genetic variations in a number of pathways, including influx and efflux transporters, significantly influence risk for cardiomyopathy, with children expressing none of the variants of concern having essentially no risk while children expressing the majority of these variants have a greater than 80% risk of developing cardiotoxicity (Visscher et al., 2012, 2013, 2015). More recently, genetic variations in Retinoic acid receptor $\gamma$ – a repair gene essential in cardiac development and remodelling and that is activated in postischemic hearts – have been found to be associated with the risk of anthracycline-induced cardiotoxicity in children on anthracycline chemotherapy (Aminkeng et al., 2015). The implication of these

**Table 1.** Genetic variations involving chemotherapy in children

| Drug class | Drugs | Genes | Impact on therapy |
|---|---|---|---|
| Thiopurines | 6-Mercaptopurine<br>6-Thioguanine<br>Azathioprine | *TMPT*<br>*NUDT15* | Myelosupression |
| Platinums | Cisplatin<br>Carboplatin<br>Oxaliplatin<br>Nedaplatin<br>Hepatplatin<br>Lobaplatin | *ERCC5*<br>*ERCC1*<br>*ERCC2*<br>*ACYP2* | Treatment resistance<br>Reduced survival<br>Shorter event-free<br>survival; neutropenia<br>Ototoxicity |
| Anthracyclines | Doxarubacin<br>Daunorubicin<br>Epirubicin<br>Idarubicin | *RARG*<br>*SLC28A3*<br>*SLC22A17*<br>*SLC22A7*<br>*UGT1A6*<br>*ABCC1*<br>*ABCC2*<br>*ABCC5*<br>*ABCB1*<br>*ABCB4*<br>*CBR* | Cardiotoxicity |
| Camptothecin and analogs | Campothecin<br>Irinotecan<br>Topotecan<br>Lurotecan | *ABCC5*<br>*ABCG1*<br>*UGT1A1*<br>*CLCO1B1* | Gastrointestinal<br>toxicity<br>Efficacy and<br>neutropenia<br>Prolonged<br>neutropenia |
| *Vinca* alkaloids | Vinblastine<br>Vincristine<br>Vindesine<br>Vinorelbine<br>Navelbine<br>Vinflunine | *CEP72*<br>*ABCC1*<br>*SLC5A7* | Peripheral<br>neuropathy |

*Source:* Modified from Elzagallaai et al. (2021).

**Table 2.** Targeted therapy for children with children

| Drug class | Indications in paediatric cancer |
|---|---|
| BCR-ABL1 tyrosine kinase inhibitor | Chronic myelogenous leukaemia in chronic phase, Philadelphia chromosome positive |
| FLT3 inhibitors | FLT3-ITD-positive acute myeloid leukaemia, de novo Acute Myelogenous Leukaemia (AML), refractory/relapsed solid tumours or leukaemia |
| CDK4 and CDK6 inhibitors | Subependymal giant cell astrocytoma Recurrent or refractory brain tumours |
| SRC family of protein-tyrosine kinase inhibitors | De novo AML |
| DOT1L histone methyltransferase (compounds under development) | Relapsed/refractory leukaemia with mixed linear leukaemia rearrangements |
| TRK inhibitor | Solid tumours that have an *NTRK* gene fusion without a drug-resistant mutation in certain TRK proteins |
| Humanised recombinant monoclonal antibody directed against CD33 | Relapsed/refractory AML |
| FGFR inhibitor | Relapsed or refractory advanced solid tumours, non-Hodgkin lymphoma or histiocytic disorders with FGFR mutations |
| EZH2 inhibitor | Epithelioid sarcoma that is locally advanced or has metastasized |
| P13K/mTOR inhibitor | Advanced cancer |
| MEK 1/2 inhibitor | Neurofibromatosis Type 1, patients aged 2 or older |
| ALK inhibitor | Paediatric patients with relapsed or refractory advanced solid tumours, non-Hodgkin lymphomas or histiocytic disorders; Newly diagnosed high-risk neuroblastoma or ganglioneuroblastoma |
| BFAF serine–threonine kinase inhibitor | Patients with relapsed or refractory advanced solid tumours, non-Hodgkin lymphoma or histiocytic disorders with BRAF V600 mutations; newly diagnosed high-grade gliomas |
| Selective inhibitor of human (ADP-ribose) polymerase enzymes | Patients with relapsed or refractory advanced solid tumours, non-Hodgkin lymphoma or histiocytic disorders with defects in DNA damage repair genes |

*Source:* Modified from Elzagallaai et al. (2021).

findings on treatment protocols for children with cancer is under active discussion.

A number of genetic variations are now known to be associated with safety and efficacy of chemotherapeutic drugs (Table 1). As the implications of these variations are factored into treatment plans, pharmacogenomics is being used to explore novel approaches to therapy, notably in the area of targeted therapy (Tokaz et al., 2022). The discovery of discrete molecular targets to enable more precise and ideally more effective therapy has accelerated rapidly over the past decade and offers the promise of reducing treatment failure and offering hope in cases of refractory disease (Table 2). It is clear that moving forward pharmacogenomics will be an increasingly important part of therapy for children with cancer.

## Neonatology

Neonates and infants are a unique subpopulation within paediatrics in that they have the most rapidly changing and developing organ function and growth. Within the category of neonates, there are 500 g, 22 week gestation preterm infants and 4,500 g large for gestational age term infants, with the difference in organ size and maturation having significant effects on drug pharmacokinetics and pharmacodynamics (Chapron et al., 2021). Most neonatal drug dosing is weight-based (mg/kg or mcg/kg), but increasing evidence confirms that there are many factors beyond size that can be taken into consideration to optimise precision drug dosing in neonates.

The ontogeny of drug-metabolising enzymes (DMEs), transporters, drug receptors and system modulators represent a very important variable in understanding precision drug therapy in neonates. Ontogeny of hepatic transporters and DMEs has been comprehensively summarised (Chapron et al., 2021), but there is a need to validate some of the in vitro work with human studies to confirm ontogeny profiles. Pharmacokinetic modelling approaches such as population pharmacokinetics (PopPK) (Rhee et al., 2022; Wu et al., 2022) and physiologically-based pharmacokinetics (PBPK) (Claassen et al., 2015; van Hoogdalem et al., 2022) have

provided many examples of improved understanding of drug disposition in neonates.

Pharmacogenetics is a potentially very important and understudied variable in drug dosing for neonates. A recent review compiles all the known publications around pharmacogenetics in neonates and infants (Yalçin et al., 2022), and the number of such investigations has indeed grown over time. Importantly, the intersection of developmental stage and genetic variation must be considered in neonates. So that, the effects of genetic variation cannot be fully appreciated until the protein of interest (DME or transport) is sufficiently expressed. The intersection of ontogeny and pharmacogenetics was demonstrated for pantoprazole clearance in neonates (Ward et al., 2010). In addition to genetic variability in DMEs, pharmacogenetic markers in drug targets and drug response modulators may also play an important role in the future of precision dosing (Elens et al., 2016; Lewis et al., 2019).

Model-informed precision dosing (MIPD) is a powerful tool to translate our understanding of drug pharmacokinetics into dosing changes at the bedside (Euteneuer et al., 2019). A major gap in pharmacokinetic knowledge implementation in neonates is the lack of translation of current well-validated PK models into improved dosing. Improved drug concentration attainment has been demonstrated in neonates for vancomycin (Frymoyer et al., 2020) and ganciclovir (Dong et al., 2018). Research teams are working towards the implementation of MIPD for drugs outside of antimicrobials as well, including precision morphine dosing in the NICU (Vinks et al., 2020).

For neonates to benefit from precision therapeutics, their advocates must engage important stakeholders to improve knowledge generation, drug development and clinical implementation of new pharmacology tools in neonates (Lewis et al., 2022). Groups such as the Paediatric Trials Network (PTN, www.pediatrictrials.org), the International Neonatal Consortium (Turner et al., 2016), drug regulation officials (McCune et al., 2017) and others are all working diligently to improve therapeutic options for the neonatal population.

## Complex and common disorders in children

The examples of the use of pharmacogenomics and Precision Medicine in both neonatal and cancer care suggest that there are many other disorders in childhood – common and uncommon – for which Precision Medicine offers the promise of more effective and safer therapy (Cohn et al., 2021). As noted above, an appreciation of the importance of drug–drug interactions is increasingly important given the common use of polypharmacy to manage complex and chronic disorders. Understanding how genetic variations can influence therapy is also a consideration. The example of TPMT is germane; while initially assessed primarily in children with leukaemia and other haematogenous malignancies, some centres now test for TPMP pre-therapy for children with rheumatological or gastrointestinal disease prior to the use of thiopurine drugs (Weitzel et al., 2018). Of interest, pharmacogenomic testing is now entering the realm of paediatric and adolescent mental health, a development that offers great promise in providing evidence to guide therapy notably given the substantial increase in the use of psychopharmacological drugs among children and adolescents (Ramsey et al., 2019; Ramsey et al., 2021).

Pharmacogenomic testing may also be useful for commonly prescribed drugs that are used for less serious conditions. Dr. Van Driest and colleagues at Vanderbilt have demonstrated that 40% of children treated with a proton pump inhibitor – among the most commonly used drugs, often for gastroesophageal reflux disease – were at a higher risk of infection related to being the CYP2C19 normal metabolizer phenotype, having double the infection rate of children who were rapid or ultrarapid metabolizers (Bernal et al., 2019).

In children and adults with suspect genetic conditions that have high genetic heterogeneity, genome sequencing is expected to become a first-tier test (i.e., a broad genetic differential diagnosis with many candidate genes or loci) instead of the second-tier technique it has typically been utilised so far. This will cut down on the time it takes to conduct several genetic tests. Genomic testing may reveal pharmacogenetic profiles, reproductive carrier status information, and genetic risk profiles for later-onset diseases. The use of genome sequencing as a preventative health tool in seemingly healthy people is uncertain at this time, but it has tremendous potential in the future.

## Challenges and opportunities in the implementation of Precision Medicine in children

A myriad of challenges is faced when preparing to implement Paediatric Precision Medicine in practice. For successful outcomes, these issues must each be considered when planning the implementation strategy, to prevent problems that could otherwise derail the process.

### Terminology: Semantic standardisation

One important, yet frequently overlooked, aspect is the need for standardisation of the chosen terminology within the respective healthcare system and agreed definitions. Many different terms are used in the literature relating to *Precision Medicine*, including *personalised medicine*, *individualised medicine, stratified medicine, precision dosing, pharmacogenetics* and *pharmacogenomics*. We advocate the use of the term *Precision Medicine*, as supported by the National Institutes of Health (Alfirevic and Pirmohamed, 2016). The advantage of this umbrella term is that it suitably encompasses the many different tools and technologies, ranging from enhanced diagnostics and disease phenotyping to targeted immunotherapies, and beyond as well as incorporating a holistic view of all factors impacting therapy, including concurrent therapy and ontogeny. Key domains of Paediatric Precision Medicine are summarised in Figure 1. Within each domain, there are numerous examples of how this may be applied in clinical practice (Capsomidis and Anderson, 2017). Some researchers expand the definition of *Precision Medicine* to incorporate *Precision Public Health*, including targeted prevention strategies, but this is beyond the scope of this paper.

The benefits of standardisation of terminologies and definitions extend beyond the clinical arena, to facilitate semantic, structural and foundational interoperability between complex data management systems. Achieving truly effective interoperability is a herculean task, given the heterogeneous nature of healthcare information systems, even within one country (Hosking, 2018; Benson and Grieve, 2021; de Mello et al., 2022). However, a harmonised approach will lay the foundations for future collaborations (including research, registries and audits), and will also facilitate the collation of agreed metrics to evaluate the implementation success across different systems.

### Financial challenges

Detailed economic considerations must be addressed at the local and regional level, to ensure that Paediatric Precision Medicine

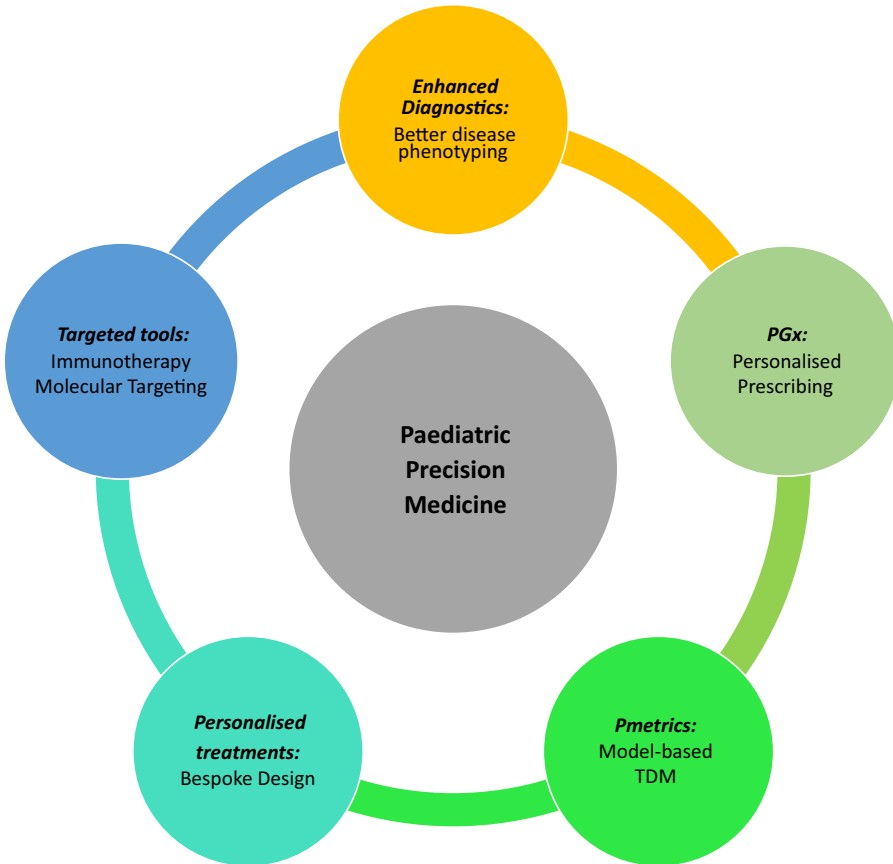

**Figure 1.** Key domains of paediatric Precision Medicine.

initiives are cost-effective for the target patient population, particularly given the current economic climate. Within the UK, this issue has been partly addressed through the funding of a nationwide Genomic Medicine Service, which includes Pharmacogenomics within its remit (Snape et al., 2019). This approach aims to ensure equity of access to the benefits of Genomic Medicine for all patients, avoiding the problems of a so-called postcode lottery, which can arise when local funding mechanisms are involved (Graley et al., 2011).

Ongoing research is needed to examine the clinical usefulness, cost-effectiveness and possible unintended future consequences of genome sequencing in healthcare systems. In certain populations, the added yield of genome sequencing over exome sequencing is modest; nevertheless, this gap will increase with improvements in data analysis and bigger data sets to compare against. The anticipation of additional clinically relevant information arising from as-yet-unexplored areas of the genome is also driving investments in genome sequencing technology. Ensuring equitable access to care informed by the DNA code, irrespective of postal code, is a challenge in many countries and needs to be a priority for policymakers.

### Susceptibility to system overload

Potential barriers to implementation also arise from the perception that modern healthcare systems frequently operate at or near their full capacity (or sometimes beyond, as is currently the case for the National Health Service) (Jefferies, 2022). This reduces the appetite of policymakers, healthcare professionals and management teams for large-scale change, such as implementing a Paediatric Precision Medicine service, which may be viewed as both costly and complicated. Therefore, careful pharmacoeconomic analysis and demonstration of clinical utility are needed to allay these fears, through demonstration that implementation will be worthwhile and will lead to meaningful benefits for patients. Even in a nationally funded model, such as that employed in the UK, there are still concerns that the mainstreaming of Pharmacogenomic (PGx) services may lead to logistical problems, where the system cannot meet the demands of the clinic, for example, if pre-emptive PGx testing became routine in clinical care (Magavern et al., 2021). Given that paediatric PGx testing is likely to be targeted (i.e., restricted to relevant patient groups) in the first instance, it is unlikely that the paediatric component alone would overload the system.

### Access, equity and transitions

An unpleasant but unavoidable fact in paediatric therapeutics is that there are massive inequities in the provision of medications to children globally, with children in high-income countries having access to a wide range of therapies while children in low-income countries struggle to have access to even the most basic and essential medications (Rieder, 2010; Bonati et al., 2021). When developing strategies for Precision Medicine, access and equity are important considerations. Even within high-income countries, access to health services and therapies is often inequitable for children in marginalised populations. An additional and important factor is that while most drug research for children is done on children of Western European and African–American descent,

most children in the world are not of Western European or African–American descent. As the population of high-income countries becomes more diverse, incorporating population appropriate PGx testing into healthcare systems will need to take diversity and cultural sensitivity into account if the benefits of Precision Medicine are to be realised not only in middle to low-income countries but also in high-income countries.

An additional issue that will need to be considered in drafting health policy and in evolving health care systems is the transition from child health care to adult care, notably for adolescents with chronic health conditions (Gray et al., 2018). This has been managed variably by different disciplines with varying degrees of success, but the addition of Precision Medicine to the mix will, while potentially improving care, provide additional challenges to ensure the optimal benefits of Precision Medicine can be achieved.

### The need to avoid a leadership void

Shared ownership of Paediatric Precision Medicine initiatives potentially brings with it the risks of leaving a leadership void, where it may feel difficult to ensure clear responsibility and accountability. Considering the example of Pharmacogenomics: this discipline arguably lies at boundary between Paediatric Clinical Pharmacology, Clinical Genetics and Paediatric Clinical Pharmacy. However, the implementation of Pharmacogenomic testing needs to be tailored to different general and subspecialty environments in Paediatrics, which will initially be in the context of children's hospitals, before consideration of primary care prescribing. The complexities of this interdisciplinary interface have likely contributed to the widespread delays in the implementation of PGx testing that have been observed to date (Lauschke and Ingelman-Sundberg, 2020). Successful implementation also relies on a seamless interface with colleagues in laboratories undertaking PGx testing and other relevant members of a multidisciplinary team (MDT) for Paediatric Precision Medicine (Figure 2; Lauschke and Ingelman-Sundberg, 2020). These political boundaries between specialties and professions can be navigated by clear signposting of the PGx leads for each team, respectively, prospective stakeholder engagement meetings and establishing clear routes of communication with the team responsible for implementing the PGx service.

### Preparing for implementation: Education and training

Appropriate workforce education and training, supported by updates to the relevant undergraduate curricula of each discipline, are key parts of the development of a successful implementation strategy. Selected examples are discussed further below.

#### Safeguarding training in interdisciplinary fields

In the interdisciplinary context of Clinical Pharmacology and Therapeutics (CPT), many relevant lessons have previously been learned within the adult CPT world (Dollery, 2006; Aronson, 2010). CPT, as a medical specialty, is unique in its holistic focus on the safe, effective and economic use of medicines (Gray et al., 2018; Turner et al., 2022). This makes CPT well-placed to support the implementation of Pharmacogenomics (and Precision Medicine more broadly) into clinical care. Yet, despite the overt need for such a discipline, and clear relevance to translational medicine, this specialty faces many challenges (Maxwell and Webb, 2006; Aronson et al., 2008). Similar issues have been reported for Paediatric CPT (PCPT) (MacLeod, 2016). Indeed, for PCPT, the situation is further complicated by challenges facing academic Paediatric Clinical Pharmacologists, and the difficulties Paediatric residents and fellows face in accessing training in research (Winch et al., 2017). Paediatric Precision Medicine services will benefit from expanding programmes dedicated to training Paediatric Clinical Pharmacologists, and sharing experiences of the implementation of such specialist services within Children's Hospitals (Bonati et al., 2006; Gazarian, 2009; Koren, 2009; Hawcutt et al., 2022). To support the infrastructure required for effective delivery of a Precision Medicine service, the multidisciplinary workforce will also need to include more clinical bioinformaticians, and laboratory-based clinical scientists, as well as focus on

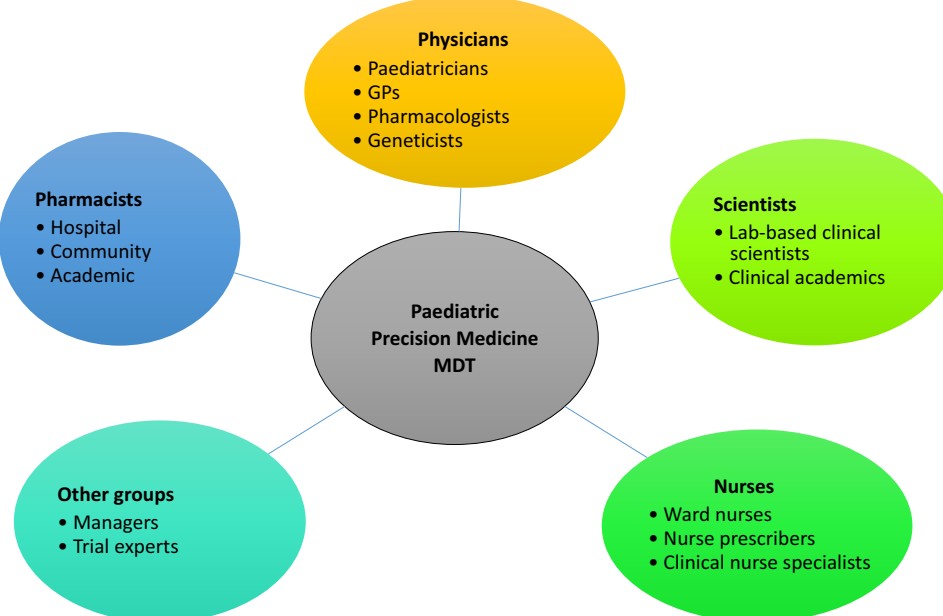

**Figure 2.** Healthcare partners for a multidisciplinary team (MDT) in paediatric Precision Medicine. Adapted from Magavern et al. (2021).

enhanced postgraduate training (Health Education England, 2021; The Academy for Healthcare Science, 2021).

### Improved lifelong therapeutics training for all paediatricians

Knowledge about the optimal use of medicines in children, including the application of *Paediatric Precision Medicine* in different patient populations, is constantly expanding; regular updates will, therefore, form an important part of Continuing Profession Development (CPD). Worldwide, all Paediatricians (both generalists and subspecialists), should have improved access to training in the principles of PCPT, which can be periodically updated throughout their careers. Physicians (or other partner healthcare professionals) with a special interest in the safe and effective use of medicines in children could then opt to undertake additional training in PCPT within the local framework for their discipline. This may take the form of so-called "credentialling," as has recently been introduced in the UK as "Special Interest Modules" available through the Royal College of Paediatrics and Child Health which can be completed after qualification as a Consultant. Specific modules in Paediatric Precision Medicine (or specific sub-domains, such as PGx would be of particular relevance.

### Improved genomics training for physicians and healthcare professionals

Limited education and training in Genomics for both doctors and allied healthcare professionals present a significant barrier to the implementation and adoption of Precision Medicine initiatives (Plunkett-Rondeau et al., 2015; Talwar et al., 2017; Nisselle et al., 2021; Schaibley et al., 2022). These educational needs can be addressed using a variety of different modalities to help satisfy the preferences of different disciplines (Slade et al., 2016; Mitchell et al., 2022). In particular, resources that are available to the whole workforce, such as "GeNotes," for example, those being developed by the NHS Health Education England Genomics Education Programme.

### The Precision Medicine MDT

Many hospitals that have introduced pre-emptive pharmacogenetic testing have reported benefits from incorporating a "*Precision Medicine Clinical Team*" within the leadership of the programme (Duarte et al., 2021). We propose that the structure of this *Precision Medicine* MDT should be adapted to suit the needs of the host institution, and recommend consideration of the development of

Precision Medicine MDT meetings (MDMs) where resources permit, if there is sufficient clinical need (Figure 2).

### Refining the model for paediatric Precision Medicine MDMs

MDMs are widely recognised to benefit healthcare delivery in many different contexts and can be associated with improved healthcare outcomes. The benefits of multidisciplinary (MDT) meetings for Genomic Medicine have previously been reported in both Clinical Genetics and related sub-specialty meetings (Taylor et al., 2019; Barker et al., 2021; Hay et al., 2022). Following the experiences of the COVID-19 pandemic, there has been a growth in the use of virtual MDMs, which can also be implemented at a regional level (Sidpra et al., 2020; Currie et al 2021; Rajasekaran et al., 2021). Although the use of virtual meetings can have its drawbacks, it may improve attendance by enabling remote participation, and some have reported that it can also improve the quality of the chairing (Luijten et al., 2021; Rajasekaran et al., 2021). The potential benefits of a regional virtual MDM approach include discussion of complex cases with Precision Medicine experts based elsewhere in a specialist centre (Hoinville et al., 2019; Sidpra et al., 2020; Barker et al., 2021). Clear case-selection criteria should be developed to justify MDM discussion, together with clear ground rules for MDMs themselves, incorporating methods to streamline the process.

### A path forward for Precision Medicine in children

The many challenges surrounding the implementation of Paediatric Precision Medicine, as outlined above, have contributed to delays and inconsistency in its application to date. However, with the healthcare technologies that are now routinely available, supported by the tools of implementation science, these obstacles can be systematically overcome, so that children can benefit from personalised therapeutics and from the growing body of evidence supporting this.

A unified approach to realising the potential of Paediatric Precision Medicine is an appealing prospect, but in reality, each healthcare system and its respective patient population is different. These differences must be acknowledged when designing the implementation strategy to ensure that the new service is suitable for meeting the needs of its patients and sustainable in the local environment.

Nevertheless, the principles of sound implementation and service delivery can be tailored for application in distinct healthcare systems, and lessons learned can be in other countries to help improve the efficiency of the process. There are implementation outcomes frameworks that can be used to guide the design process (Table 3).

**Table 3.** An implementation outcomes framework applied to paediatric Precision Medicine, focusing on the paradigm of pharmacogenomic testing

| | |
|---|---|
| Acceptability | Stakeholder engagement meetings to ensure the acceptability of PGx testing proposals to healthcare professionals, parents, children and young people |
| Adoption | Agreement from national and local healthcare management teams to fund PGx testing with the development of adoption plan and implementation strategy |
| Appropriateness | Interprofessional agreement from relevant healthcare professionals regarding the clinical utility of proposed PGx testing plan |
| Feasibility | Use of pharmacoeconomic modelling to gauge affordability of plans and regional/local pilot studies to demonstrate feasibility at a local level |
| Fidelity | Guidelines regarding patient selection to ensure only used in clinically appropriate contexts, with validation rules in EHR where feasible |
| Costs | Agreement between hospital, primary care providers, laboratories, and, if relevant, insurers regarding costs and reimbursement |
| Coverage | Monitoring to ascertain how many patients are offered (and accept) PGx testing and ensure equitable access across the target patient population |
| Sustainability | Planning for PGx testing to remain routinely available and future proofing (e.g., planning for periodic re-analysis of PGx data where feasible) and planned education programme |

## Consideration of children within the design phase

Ensuring that the needs of children are considered during the initial system-wide planning phase for Precision Medicine strategies should be a core goal whenever this is possible. During the large-scale planning processes, there is a risk that children and paediatric healthcare may be included only as an afterthought, and this leads to unnecessary delays in paediatric implementation strategies. For example, the recent report of the Royal College of Physicians and British Pharmacological Society, "*Personalised prescribing: using pharmacogenomics to improve patient outcomes,*" contains no specific section dedicated to Paediatrics, although promisingly the working party did at least hear a presentation from an expert representative from the RCPCH.

## Embedding implementation into research programmes

Implementation strategies should ideally be planned during the primary research (or preferably beforehand) so that engagement with relevant stakeholders can take place while a research study is running. To maximise engagement with Paediatric Precision Medicine, implementation planning will entail discussions with all stakeholders, including members of the multidisciplinary healthcare team, professionals in bioinformatics and healthcare technology, patient groups, managers and representatives from the industry (Figure 2). In paediatrics, engagement with children and young people has been recognised as an important aspect of paediatric research studies and helps to ensure that studies include endpoints that are meaningful for patients and that the *voice of the child* is always heard (Rieder and Hawcutt, 2016).

## Embedding research into implementation

Conversely, the routine incorporation of research into implementation strategies is another route to improve the effectiveness of Paediatric Precision Medicine in practice. Every patient should have the right to be involved in state-of-the-art research studies relevant to their condition. Indeed, the field of Paediatric Oncology has led the way in bringing research to the clinical frontline, demonstrating that enrolment in a clinical trial can become the standard of care (Unguru, 2011). This helps to ensure that patient involvement in high-quality multicentre studies becomes an intrinsic part of routine clinical practice. Furthermore, when clinically appropriate, Precision Medicine research studies can be designed to recruit children and adults simultaneously, if this is scientifically and clinically sound; the feasibility of this approach has previously been demonstrated (Lonsdale et al., 2020). It is ethically important to include children in research where there is a clinical need, but to avoid unnecessary studies at the same time (Hoppu, 2009).

**Open peer review.** To view the open peer review materials for this article, please visit http://doi.org/10.1017/pcm.2022.14.

**Author contributions.** All authors have contributed to this manuscript.

**Financial support.** C.B. is funded by the UK National Institute for Health Research (NIHR ACF-2019-17-004). M.R. holds the Canadian Institutes of Health Research-Glaxo Smith Kline Chair in Paediatric Clinical Pharmacology at the University of Western Ontario.

**Competing interest.** The authors have no competing interests to declare.

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
