## [Reviewer Report]

*Comments to Author*: This is a useful strategic overview of the history and current situation for precision medicine in children.. i am using the page numbers in the top left of each page for reference.

Major points:

1) This is more of an observation than a request for change - it is a very "person centric" review, featuring and name checking many of the key figures who established the scientific disciplines on which the discoveries were based. - could the title reflect this? "Precision medciine in paediatrics past and present" or somesuch??

2) There is a real risk that precision medicine will exacerbate health inequalities - somthing that children are particularly prone to experience. in the later sections I think this deserves a mention (either in a global health perspecitve, or access to healthcare)

Minor points:

1) Page 3 (introduction) paragraph 3 - last sentence - I agree, but there has also been a shift in ethical opinions (see Nuffield council of bioethics reports for example) that have helped - in additoin to advocacy, that drove regulatory changes.

2) Paragraph 5 page 3 - ""received either one or not prescriptions" - unclear (i get it but it could be polished)

3) It;s a personal quirk - but unoffical titles ("father of american pediatrics" - page 3, last paragraph) dont add anything.

4) Page 4, paragraph 2 - I would add illicit drugs 9(given with or withour therapeutic intent to the list , of "traditional or folk" medicines. Mostly CBD

5) page 5, paragraph 3 - I would add consent to the 3 processes

Page 6 - "The Neonate" - the previous title was "Oncology" - so should this be "Neonatology" to match??

---

## [Editor Report]

*Comments to Author*: Your manuscript was reviewed by me and an external reviewer. We both enjoyed reading your article and would like to proceed with publication, after a few minor points are addressed. Comments from myself are below. We look forward to receiving a revised manuscript.

1. Please consider commenting about ‘adolescent transition’ based on precision medicine.

2. Please check Table 2 ‘Targeted Therapy for Children with Children’; for example BCR-ABL1 (not Ber-Abl); to my knowledge, MTOR inhibitors are not used to treat FLT3-ITD AML (whereas FLT3 inhibitors are); I don't know of an approved DOT1L inhibitor; MEK1/2 (has autocorrected to a half symbol).

3. Please define ‘PGx’ when first appeared in the text.

4. Please reconsider including TPMT polymorphism as a risk factor for platinum-induced otoxicity; this association is not biologically coherant (the enzyme has no known role in platinum metabolism) and the association has not been consistently found (in fact one of the articles cited, Clemens et al, did not find an association). Overall, a genuine associations seems unlikely (despite is inclusion in an FDA database).

5. The manuscript would benefit from a final read for grammar/typos.

---

## [Reviewer Report]

*Comments to Author*: The authors have considered the reviewers comments and adapted the manuscript accordingly - I am very happy with it